# Contrasting carbon cycle along tropical forest aridity gradients in West Africa and Amazonia

Huanyuan Zhang-Zheng [1,2] ✉, Stephen Adu-Bredu[3,4], Akwasi Duah-Gyamfi [3], Sam Moore[1], Shalom D. Addo-Danso[3], Lucy Amissah [3], Riccardo Valentini [5], Gloria Djagbletey[3], Kelvin Anim-Adjei[3], John Quansah[3], Bernice Sarpong[3], Kennedy Owusu-Afriyie[3], Agne Gvozdevaite[1], Minxue Tang [6], Maria C. Ruiz-Jaen [7], Forzia Ibrahim[8], Cécile A. J. Girardin[1], Sami Rifai[9], Cecilia A. L. Dahlsjö [1], Terhi Riutta[1], Xiongjie Deng [1], Yuheng Sun[10], Iain Colin Prentice [6,11], Imma Oliveras Menor [1,12] & Yadvinder Malhi [1,2] ✉

Tropical forests cover large areas of equatorial Africa and play a substantial role in the global carbon cycle. However, there has been a lack of biometric measurements to understand the forests' gross and net primary productivity (GPP, NPP) and their allocation. Here we present a detailed field assessment of the carbon budget of multiple forest sites in Africa, by monitoring 14 one-hectare plots along an aridity gradient in Ghana, West Africa. When compared with an equivalent aridity gradient in Amazonia, the studied West African forests generally had higher productivity and lower carbon use efficiency (CUE). The West African aridity gradient consistently shows the highest NPP, CUE, GPP, and autotrophic respiration at a medium-aridity site, Bobiri. Notably, NPP and GPP of the site are the highest yet reported anywhere for intact forests. Widely used data products substantially underestimate productivity when compared to biometric measurements in Amazonia and Africa. Our analysis suggests that the high productivity of the African forests is linked to their large GPP allocation to canopy and semi-deciduous characteristics.

As the most productive terrestrial ecosystems, tropical forests and savannas account for over 60% of global terrestrial gross primary productivity (GPP) and feature large spatial variation[1,2]. Previous studies suggest that many African forests may have carbon budgets and dynamics that are different to those of Amazonian forests[3]. This difference is likely because the two regions have experienced very different biogeographic and climatic histories and have different current environmental exposures[3]. For instance, a satellite-based study suggested that the NPP trend of African forests during past decades has contrasted with that of Amazonian forests[4]. Furthermore, a stable and positive carbon sink into above-ground biomass for African forests was reported to contrast with the declining trend in Amazonian forests, owing to the different mortality rates following drought events[5]. A study focusing on the decadal long-term drying trend across West Africa also reported increases in forest biomass accompanied by a relative increase in deciduous species[6]. Bennett et al.[7] found that African forests were more resistant to drought than other tropical forests, likely linked to the long history of greater climatic variation in the African tropics, compared to Amazonia[3]. Moreover, African tropical forests generally boast a higher biomass stock than neotropical forests[8,9]. Field measurements of NPP[10] revealed that along a wet-dry gradient in West Africa there was evidence that NPP was higher compared to equivalent gradients in Amazonia[11].

Despite their importance for the global carbon cycle, our confidence in tropical primary productivity estimation, especially for African tropical forests, remains low[12–15]. The large uncertainty is an inevitable consequence of the lack of field evidence, which has been identified as a common issue in the carbon cycle community[16–18]. For example, GPP can be estimated by non-biometric methods, such as eddy covariance tower measurements and remote sensing. However, in the entire African tropical forest region, there was only one eddy covariance tower with published results (in Ankasa, Ghana), reporting three years of GPP (2011–2014)[19,20]. Satellite-based GPP products (e.g. MODIS GPP) struggle to provide reliable estimates of GPP of the region owing to the dense cloud cover, which extends across both wet and dry seasons[21–23]. Further, satellite-based GPP products also require vegetation process modelling that needs to be informed by field measurements. Unlike GPP, autotrophic respiration and allocation of GPP rely on intensive field studies for quantification. To date, there has been no field assessment of GPP, Ra, GPP allocation and carbon use efficiency (CUE, the fraction of GPP allocated to NPP) of any African forests. As most vegetation models calculate net primary production (NPP) and biomass from GPP and autotrophic respiration[24], the dearth of these field measurements leads to simplified model assumptions or disagreement among models, restricting our ability to predict the effects of global change on the carbon dynamics of the biosphere[25–29].

In this work, we present a comprehensive carbon budget for forest sites in Africa and compare these to previously published results for Amazonian[11] (Fig. 1). This dataset comprises 14 one-hectare plots spanning an aridity gradient and six years of field data collection, including measurements of autotrophic and heterotrophic respiration that enable integrated assessment of GPP and its partitioning. Specifically, we ask: (1) is the higher NPP reported for the West African forests also reflected in higher GPP, or higher CUE? (2) Is the relationship between GPP and aridity similar along the West African and Amazonian forest gradients? (3) Do the studied West African and Amazonian forests share similar carbon flux partitioning patterns? (4) How do our in-situ measurements of tropical forest productivity compare with widely used global data products (MODIS and FLUX-COM – an eddy covariance tower based product)?

## Results and discussion

### Comparison between Amazonia and West Africa

Our comparison reinforces that the carbon flux of the studied West African forests is distinctly different to that previously reported for Amazonian tropical forests. Overall, the total GPP and autotrophic respiration were higher in West Africa than in Amazonia along equivalent aridity gradients (Fig. 2a) (see statistics in Table S2), albeit the wettest sites share similar values. The difference arises mostly from differences in canopy NPP and leaf respiration (Fig. 2g, h). The high leaf respiration of West African forests is based on elevated dark respiration measurements, not on higher leaf area index (LAI) (Table S1), consistent with previously reported high net assimilation rate and dark respiration of West African species[30–33]. In contrast, stem woody productivity of the wettest sites and rhizosphere respiration appear higher in the Amazonian sites (Figs. 2i, l), implying very different allocation patterns between the two studied regions (Fig. 3). Overall, these differences lead to our estimates of GPP being much higher in the West African gradient than in Amazonia, particularly in the drier sites. However, unlike biometric estimates, FLUXCOM (climate-based extrapolation from global flux tower networks) and MODIS (satellite remote sensing and vegetation modelling) estimated West African forests to have lower GPP than Amazonian ones (Fig. 4). This is also evident in previous studies presenting other vegetation models or satellite-based products[1,34,35]. We found FLUXCOM and MODIS consistently underestimated both Amazonia and West African forests GPP (Fig. 4), a finding consistent with previous Amazonia data-model comparison studies[36,37]. There was only one eddy covariance tower for

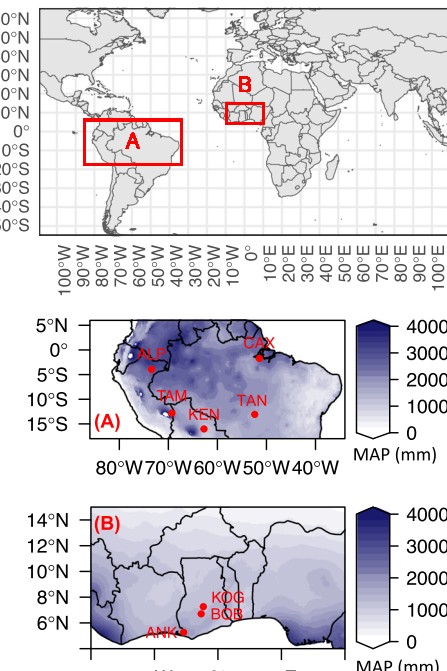

**Fig. 1 | Map of the study sites.** A Amazonia aridity gradients and B West African aridity gradient. Colour scale illustrates mean annual precipitation (MAP). Each red dot denotes a site. Each site contains multiple one-hectare plots (Table S1).

West African forests, situated at one of our study sites Ankasa (ANK)[19,20]. Previous studies found that, at this site, the flux-tower GPP (yearly mean varying from 22 to 36 MgC/ha/year) is larger than vegetation model simulation[38–40], but both are substantially smaller than in-situ biometric measurements (40.1 MgC/ha/year)[38–40]. The above synthesis reveals an acute data-model discrepancy in tropical forest productivity estimates, especially for West African forests, which require more detailed investigation.

### Variation within sites and along the aridity gradients

It is striking to find that NPP is generally maintained, or even enhanced, in seasonally dry forests compared to wet forests before dropping off in very dry forests in both Amazonia and West Africa (Fig. 2c). Along both transects, CUE peaks in the mid-aridity plots but this peak appears to occur for different reasons in Amazonia and Wear Africa. In the Amazonian sites, leaf production is not higher, but leaf respiration is lowest at intermediate aridity sites leading to the high CUE. In the West African sites, leaf production is so high at the mid-aridity site that drives the high CUE at the mid-aridity site, even though leaf respiration also increases. Woody stem productivity does show strong site-to-site variation, but no clear pattern along both aridity gradients. Surprisingly, along the African transect, GPP, autotrophic respiration, CUE and NPP are highest in the mid-aridity site. In contrast, in the Amazonian sites, GPP and autotrophic respiration decrease toward dry sites. In short, NPP follows the pattern of CUE and GPP in Africa but follows CUE instead of GPP in Amazonia. The contrasting patterns of GPP along both aridity gradients are not simulated by MODIS nor FLUXCOM (Fig. 4). MODIS simulates that tropical GPP should peak at mid-aridity sites, while FLUXCOM predicts that tropical GPP decreases steadily toward drier sites. This study therefore highlights that semi-deciduous forests with intermediate aridity can be more productive than wet evergreen forests, as seen in comparing the mid-aridity site (BOB) to wet sites (ANK and ALP) (Fig. 2a, c).

Within-site variation (i.e. variation between adjacent plots in a site) can reveal the impact of local environmental factors. The seasonally flooded swamp forest ANK-03, despite having less stem

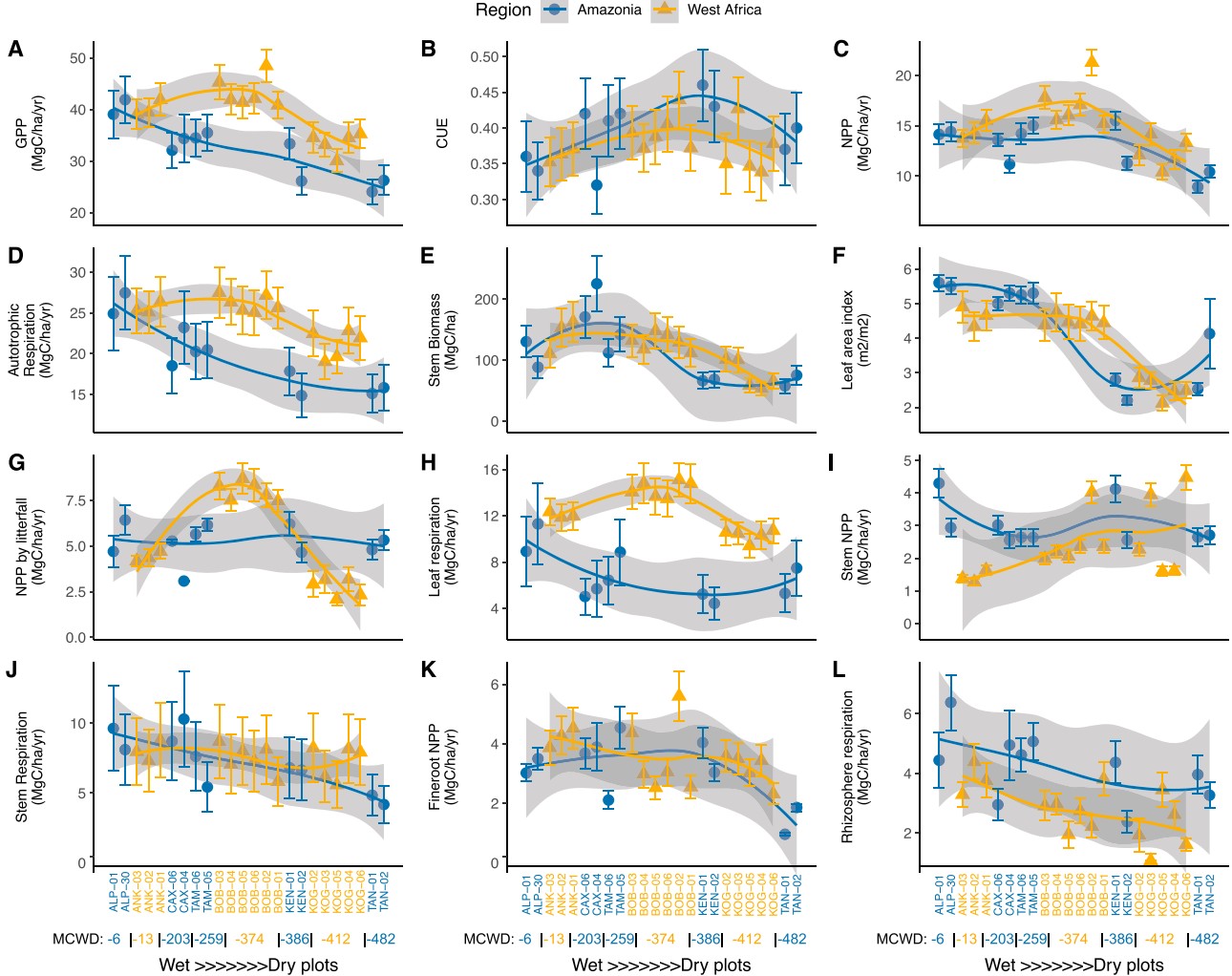

**Fig. 2 | Biometric estimates of various components of carbon fluxes.** Both Amazonia plots (blue dot) and West African plots (yellow triangle). Local polynomial regression fitting lines (LOESS) are drawn for each region. Each dot represents one one-hectare plot. The error bars represent the uncertainty of estimates (see Supplementary Method for uncertainty estimation). The x-axis is a factorial order of aridity. From wettest (left) to driest (right), sites were ranked by maximum climatological water deficit (MCWD mm year⁻¹). Within sites, plots not separable by MCWD were ranked by in situ measured surface soil moisture, which reflects local soil moisture retention and under canopy humidity (Table S1). The curves and their uncertainties (grey zone) were drawn by local polynomial regression for illustration purposes only; they do not contain relevant statistical information. Source data are provided as a Source Data file. The figure shows **A** net primary productivity (NPP), **B** gross primary productivity (GPP), **C** carbon use efficiency (CUE = NPP / GPP), **D** autotrophic respiration, **E** stem biomass, **F** leaf area index, **G** NPP by litterfall **H** leaf respiration **I** stem NPP **J** stem respiration **K** fineroot NPP and **L** rhizosphere respiration. See Supplementary Data 1 for the definition of each carbon flux component.

biomass (135 MgC/ha) than the adjacent dry substrate forest (163 and 153 MgC/ha for ANK01 and ANK02), does not have significantly different NPP or GPP (Fig. S4). Similarly, the different history of logging in Bobiri (see Methods) does not seem to result in significantly different NPP or GPP (except BOB02, Fig. S4). The steep variation of biomass along the dry forest-savanna transition in Kogyae (KOG) leads to variation in NPP but not GPP. Similarly, GPP and NPP vary considerably along a continental aridity gradient in Amazonia but variation amongst plots within the same site is mild[11,41–43]. Plot-to-plot variation of NPP with logging intensity was reported in Borneo[44], but that was for much higher intensities of logging than at the Bobiri site. Such plots-to-plot variation on GPP and NPP were not seen for our Bobiri study and another West African study on logging[45]. The above implies that aridity, instead of local edaphic or other environmental factors, is the primary driver of productivity variation of West African forests. The lack of plot-to-plot variation within a site also increases confidence in the exceptionally high GPP measurements found at Bobiri.

## Contrasting patterns in carbon allocation

CUE in African forest sites appeared generally lower than CUE in Amazonian forests (Fig. 2b). Given that the Amazonia CUE is lower than the global average, our finding further expands the global range of CUE and confirms that, globally, mature tropical forests are low CUE ecosystems[46–50]. Among all plots, CUE has no correlation with GPP (Fig. S1), whereas both GPP and CUE have a significant correlation with NPP. Overall, CUE explains less spatial variation in NPP than GPP, but CUE does exhibit considerable spatial variation and assuming fixed CUE parameters in a model (e.g.[51]) for both studied gradients would misrepresent the spatial variation of NPP[47].

Further differences between Amazonian and West African forest carbon cycling were revealed in an investigation of photosynthate allocation into canopy, fine root, and woody NPP and respiration (see Fig. 3 for definition). Note that this is the partitioning of productivity and metabolic activity instead of the more commonly reported partitioning of biomass[52]. In both regions, the allocation pattern of autotrophic respiration is more homogeneous across plots than the

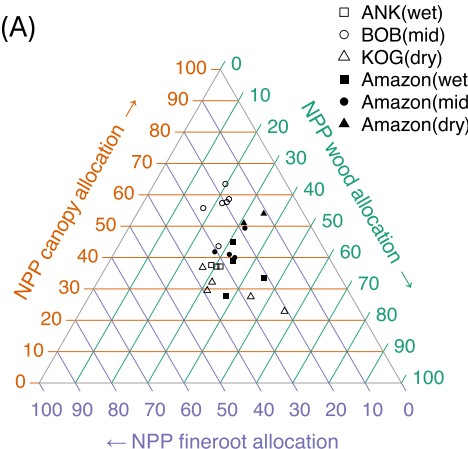

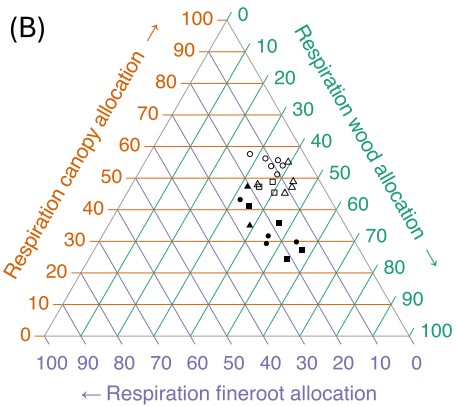

**Fig. 3 | Forests carbon partitioning.** The figure shows the partitioning of **A** net primary productivity (NPP) and **B** autotrophic respiration. One dot represents one plot, for West African (hollow marker) and Amazonia (solid marker). The aridity of plots was illustrated by marker shape. Allocation percentage should be read by following the arrow and ticks on each axis. For example, in the NPP diagram, the top hollow-round dot (BOB05) represents high allocation (63%) to canopy (red), 20% to wood (green), and 17% to fine roots (blue). The percentage of allocation was calculated as follows: NPP canopy allocation (%) * NPP = NPP_fine_litter_fall + NPP_herbivory; NPP wood allocation (%) * NPP = NPP_all_stem + NPP_coarseroot + NPP_branch; NPP fineroots allocation (%) * NPP = NPP_fineroot; Respiration canopy allocation (%) * R_autotrophic = R_leaf; Respiration wood allocation (%) * R_autotrophic = R_stem+R_coarse_root; Respiration fine roots allocation (%) * R_autotrophic = R_fine_root. See Methods for the definition of each component. Source data are provided as a Source Data file.

allocation of NPP (i.e., points in Fig. 3b are more clustered than in Fig. 3a). For NPP, the Ankasa wet rainforest site shows allocation patterns within the ranges of Amazonian plots. The mid-aridity site (Bobiri) consistently allocates more to the canopy than the reported Amazonian sites do. In contrast, the dry Kogyae site allocates consistently less to canopy and more to wood. For autotrophic respiration, leaf respiration clearly separates West African plots from Amazonian plots (Fig. 3, Fig. 2h). The dry Kogyae forests allocate more to wood respiration, and the mid-aridity Bobiri sites allocate more to the canopy, both of which match the NPP allocation trends. Overall, the carbon allocation of tropical forests to different organs is highly variable between sites, which complicates aspirations for relatively simple modelling of allocation. For example, respiration allocation to canopy ranges from 24% to 63% and NPP allocation to canopy from 23% to 63%,

values substantially higher and more spatially dynamic than extra-tropical forests[53,54]. Both NPP and respiration allocation to woody components also vary considerably in this study, from around 20% to 60%, while allocation to fine roots varies less.

The high variability in allocation patterns raises questions about the "fixed-ratio method" for intact forests carbon modelling[55,56]. Dynamic NPP allocation to some vegetation components is considered by current models but with models realising drastically different patterns[29]. Despite the large spatial variation, on average, tropical NPP partitioning to the canopy (42.1%) is slightly larger than that to wood (31.1%) (Table S3), a pattern not captured by satellite-based products or vegetation models which substantially underestimates allocation to canopy[51,54]. Moreover, along both the Amazonia and West African aridity gradients, spatial patterns in stem NPP are poor analogues for the spatial pattern of total NPP or GPP – hence inferences on tropical forest productivity based on forest censuses (stem diameter) alone should be treated with caution (Figs. 2i, a). Previous studies also found canopy NPP could better explain spatial variation of total NPP than stem NPP[57,58]. The gathered evidence highlights the great importance of CUE and canopy NPP, whereas previous literature has paid much more attention to GPP and stem NPP[59]; hence, future research is needed to understand the mechanistic principles underlying carbon use and allocation in tropical forests ecosystems.

## Why are West African seasonally dry forests so productive?

As the wet evergreen forests in Amazonia Allpahuayo (ALP) and West African Ankasa (ANK) show similar levels of productivity (Figs. 2a, c), the reasons for the mean high productivity in the West African transect resides with the drier African plots (BOB and KOG). The most productive plot, BOB02, has a very high estimated GPP at 48.5 ± 3.2 and NPP at 21.3 ± 1.29 MgC/ha/year, the largest values reported this far for natural forests stands to our knowledge[16] (although it should be noted that farmland or logged forest may have even higher NPP[44,45]). BOB02, compared to other plots in this study, has exceptionally high GPP, CUE, NPP, and autotrophic respiration simultaneously. All six plots in the Bobiri (BOB) Forest reserve (54 km²) show generally very high GPP (Fig. S2) through multiple years of measurements, so this high productivity is representative of the wider region. High NPP was also measured at another two one-hectare plots in old-growth forests at Kakum, approximately 200 km to the south of Bobiri (BOB)[45] and by a previous study focusing on aboveground NPP[60].

Since the most distinctive difference between Bobiri forests and Amazonian forests is in carbon allocation to canopy (high leaf NPP and respiration), the high productivity of this mid-aridity site should be associated with its special semi-deciduous leaf phenology - remaining green all seasons but with only 5.2 ± 0.6 months leaf lifespan[10], substantially shorter than Amazonia (>12 months)[61]. In other words, Bobiri has similar LAI and leaf biomass to Amazonian sites but different leaf turnover time, considering NPP = biomass x turnover. Such a carbon strategy, however, entails higher per leaf area photosynthesis to facilitate rapid leaf growth and turnover[62]. Species at BOB and KOG are indeed characterised by high photosynthesis rates[30,31], for example, *Triplochiton scleroxylon* (BOB), *Nesogordonia papaverifera* (BOB), *Afzelia Africana* (KOG) *and Pterocarpus erinaceus* (KOG), all deciduous species widespread across West Africa (Table S4). Along our West African aridity gradient, as indicated by photosynthetic traits measurements (Table S1), photosynthesis rate per leaf area and solar radiation increases toward drier sites[63,64] while LAI decreases (Fig. 2f), resulting in the mid-aridity site being the most productive. In other words, light use efficiency increases toward drier sites, but the fraction of absorbed photosynthetically active radiation (fAPAR) decreases. Similar to BOB02, another relatively fertile mid-aridity plot in Bolivian Amazonia, Kenia (KEN01), with the highest NPP along the Amazon aridity gradient, also featured a similar carbon strategy with more canopy seasonality, higher CUE, and quicker carbon turnover (see

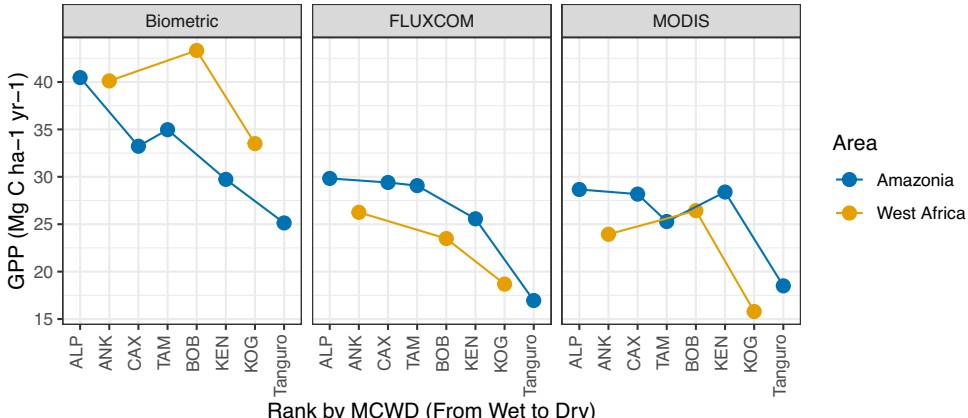

**Fig. 4 | Comparison of gross primary productivity (GPP, MgC ha-1 year-1) estimates from various sources.** The figure shows GPP for five sites in Amazonia (blue dots) and 3 sites in West Africa (yellow dots) by FLUXCOM, in situ biometric methods, and MODIS. One dot represents one site; each site contains multiple one-hectare plots (Table S1). For in situ biometric GPP, a mean value is calculated across plots. Source data are provided as a Source Data file.

details of Kenia site[42] and its leaf lifespan and seasonality[61,65]. Both Bobiri and Kenia are relatively fertile sites with high pH soils – such fertility may be requisite to support leaf construction costs and rapid leaf turnover. It is possible that BOB02 and KEN01 are recovering from previous disturbances (see Methods)[42,66,67] or that it is simply a common characteristic of semi-arid forests that experience higher rates of natural disturbance (drought or fire) and are dominated by fast-growing light-demanding species[68]. The high photosynthetic rate is consistent with short leaf lifespan and semi-deciduous phenology[69,70]. Nonetheless leaf traits provide the proximate factor explaining high productivity in seasonal African forests, not the underlying reason. This underlying reason is likely related to a combination of relatively high fertility (and associated cheap leaf construction costs) and seasonality favouring deciduousness.

Our findings suggest that semi-deciduous sites might be the most productive tropical forests, even more than wet evergreen forests. This counterintuitive finding is, in fact, consistent with ecological theories; In mature evergreen forests dominated by gigantic trees (see number of trees in Table S1), each tree maximises its own resource acquisition leading to an increasingly conservative community that prioritises defence and longevity over rapid growth, leading to lower overall community resource use efficiency and lower primary productivity[71–73]. In contrast, drier forests experiencing frequent dry spells are more subject to occasional disturbance. As predicted by the intermediate disturbance hypothesis, the site may have high productivity and turnover rate[42,74]. Previous studies reveal that long-term drying trends have driven West African semi-arid forests toward increasing deciduousness[6,75], perhaps making them more resilient to climate change than wet-evergreen Amazonia forests[5,7]. In the longer-term millennial timescale, the fluctuating climate of West Africa has resulted in a dynamic forest-savanna transition that has migrated and forth over the Holocene[76–78]. Forest species that are adaptable and able to grow rapidly when opportunities arise, likely have thrived under this variable climate compared to the less variable climate of Amazonia. Thus, our analysis highlights that the studied West African forests have a contrasting carbon strategy to the studied Amazonian forests. However, as our African sites are restricted to one country, more evidence is required to achieve a more geographically distributed understanding of the African forest carbon cycle (and indeed of the Amazonian forest carbon cycle).

In summary, we present a complete carbon budget of African tropical forests by utilising extensive field measurements across 14 one-ha plots, contributing towards a more comprehensive understanding of forest carbon allocation worldwide. The findings reveal

that the productivity of West African seasonally dry forests could exceed that of Amazonian lowland forests but has been previously underestimated by satellite and model products. Furthermore, the study features the discovery of the world's most productive forests measured to date – the Bobiri forest. However, it should be noted that the Bobiri forest is now a mere 54 km² patch. The West African tropical forests are severely fragmented[79] and increasingly face land use change pressures[80,81]. These forests not only present significant carbon stores[8,9] and high productivity (Figs. 2e, a), but are also potentially adaptable and resilient to climate change[6,82]. These irreplaceable and highly productive forests merit conservation and restoration attention to avoid continued carbon loss, and for a multitude of benefits they provide to the local environment, biodiversity and society.

## Methods
### The Ghanaian aridity gradient
As part of the Global Ecosystems Monitoring (GEM) network[83], 14 one-hectare plots were established within three forest reserves in Ghana (Table S1, Fig. 1) spanning an aridity gradient. There are three plots in the wettest forest reserve, Ankasa National Park (ANK), which receives a mean annual precipitation of 2050 mm. The medium-aridity forest reserve Bobiri (BOB) hosts six plots with a rainfall of 1500 mm. At the driest end, Kogyae Wildlife Reserve (KOG) has five plots with a rainfall of 1200 mm. These plots have a very similar mean annual temperature, but span a steep precipitation gradient, which provided a "natural laboratory" to investigate the effects of aridity on forest productivity and respiration. The aridity of each plot is indicated by maximum climatological water deficit (MCWD mm/year) or, if not discernible by MCWD within a site, by in situ measured surface soil moisture (Table S1), in accordance with soil hydrology modelling of these sites[64]. Along the aridity gradient, rainfall seasonality (Table S1), vegetation seasonality and deciduousness increased considerably toward the drier sites[10,61,65], whereas LAI and tree density decreased toward the drier sites[64]. More information about the study sites, soil properties, hydrology and climate regime can be found at[10,30,84,85]. Forests in ANK and BOB have never burnt (to our knowledge), but KOG experiences wildfire roughly every decade[19,86]. As the study sites are situated in different forest zones based on specific endemism, we believe that ANK BOB and KOG are representative study sites of West African forests but they may not represent other African regions[87].

Although plots within the same forest reserve share very similar air temperature and precipitation, they differ dramatically in terms of soil moisture and composition of the vegetation community because of their soil properties, topography, and disturbance history. ANK is a humid rainforest and Pleistocene refuge (i.e. persisted as rainforest

during arid glacial periods) with three plots spanning dry uplands (ANK01 and ANK02) and seasonally inundated riverine lowland (ANK03). BOB is a semi-deciduous forest where the plots span a gradient with selective logging history, ranging from the intact forest (BOB01) to forest lightly logged (2-3 stems/ha extracted) in 1959 (BOB02 – 04) and 2001 (BOB05 –06). KOG is a forest-mesic savanna transition where local soil factors influence vegetation type from dry forest (KOG02 and 03) to savanna (KOG05 and 06)[64]. Within any site (e.g., within ANK), there are common species across plots, but the most abundant species may still differ; there are almost no common species across sites.

## Detailed carbon budget quantification

The study quantified the whole carbon cycle of 14 West African plots (one-hectare each) with a bottom-up method mostly following the GEM protocol[83,88]. For each GPP component, a definition and brief description of the sampling method is provided in (Supplementary Data 1). The detailed sampling technique, calculation and scaling process, and references are explained in Supplementary Method. For completeness and consistency, we sourced the estimate of some NPP components from a previous analysis of the same dataset[10], but all NPP values are presented again in this manuscript as Source data. Carbon allocation (the partitioning of GPP) was illustrated with a ternary plot (Fig. 3), drawn by the R package "Ternary." The 'z-test' was used to assess the difference among plots (Fig. S4).

Our reported NPP and R_autotrophic are completely independent measurements. The plot-to-plot variations (Fig. 2) in NPP and R_autotrophic are highly similar, and analogous allocation patterns (Fig. 3) could also be found between NPP and R_autotrophic, which reinforces the reliability of the findings. Furthermore, we found the sum of R_soil_heterotrophic and R_cwd was roughly equal to the sum of D_cwd, D_litter_fall, and D_root in each site (Figure S3). This is a valuable cross-check to validate our carbon flux measurements because R and D are independent measurements, and they are expected to be equal in steady-state conditions of little net soil carbon accumulation.

The study also featured a comparison with detailed carbon budgets of Amazonian plots where the same sampling protocol was applied[83]. The carbon budget of these plots is reported by[11]. The western Amazonian sites (on relatively fertile soils) include Allpahuayo in NE Peru with almost no seasonality[89], Tambopata in SE Peru with a moderate dry season[90] and Kenia in Bolivia featuring a strong dry season[42], which is situated at the transition between humid Amazon forest and *chiquitano* dry forest. The eastern Amazonian sites (on relatively infertile soils) include Caxiuanã, humid forests in NE Brazilian Amazonia[91] and Tanguro, dry forests in SE Brazilian Amazonia[68], which sit close to the dry forest-savanna ecotone. Both Amazonia and West African aridity gradients show increasing seasonality toward dry sites. Since this paper focuses on the spatial variation of the carbon budget, not seasonal variation (presented here[65]), we average monthly measurements to an 'annual mean' for both study gradients. See Table S1 and references above for more characteristics of the sites.

## FLUXCOM and MODIS

Using Google Earth Engine, we retrieved MODIS GPP from the collection MOD17A3HGF during 2001 to 2020. This collection of GPP has been cloud contamination filtered and gap filled by the data providers. We chose the 'RS_METEO' version of FLUXCOM because the magnitude of GPP in this version does not involve uncertainty from MODIS FAPAR, which makes the comparison between FLUXCOM and MODIS GPP more independent. We extracted GPP of the studied plots using their coordinates and calculated the mean annual value per site.

## Data availability

Data generated in this study have been deposited in the 'figshare' database under accession code https://doi.org/10.6084/m9.figshare.23615472. The full set of carbon budget data, beyond what has been presented in this paper, is also supplied. Plots of environmental information and field photos are provided in Supplementary. Source data are provided with this paper.

## Code availability

Code used in this study have been deposited in the 'figshare' database under accession code https://doi.org/10.6084/m9.figshare.23615472.

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

## Acknowledgements

We thank Roberto Salguero-Gómez, Simon Lewis, Jesus Aguirre Gutierrez, and David Bauman for valuable suggestions. We thank Armel T. Mbou, Agnes de Grandcourt and Giacomo Nicolini for their help with fieldwork in Ankasa. Y.M. is supported by the Jackson Foundation and the Leverhulme Trust. H.Z. received the Henfrey Scholarship (by St Catherine's College, Oxford) and the Tang Scholarship (by the China-Oxford Scholarship Fund). This work is a product of the Global Ecosystems Monitoring (GEM) network (gem.tropicalforests.ox.ac.uk). Fieldwork was funded by grants from the UK Natural Environment Research Council (NE/I014705/1 and NE/P001092/1), European Research Council Advanced Investigator Grants to Y.M. (GEM-TRAITS: 321131) and R.V. (Africa-GHG: 247349) under the European Union's Seventh Framework Programme (FP7/2007-2013), and an Africa Oxford Initiative Catalyst Grant to H.Z. We also acknowledge the Wildlife Division of the Forestry Commission in Ghana as well as the many field assistants who helped with data collection from the field.

## Author contributions

H.Z-Z. contributed to conceptualisation, formal analysis, investigation, and writing. S.A.B. and S.M. contributed to conceptualization, investigation, and resources. A.D-G., L.A., S.D.A-D., F.I., R.V., G.D., K.A-A., K.O-

A., A.G., M.C.R-J., J.Q. and B.S. contributed to investigation, resources, and data collections. C.A.J.G. contributed to investigation, formal analysis, and writing. S.R., C.D., X.D., T.R., M.T. and Y.S. contributed to formal analysis and data curation. I.C.P. contributed to supervision and writing. I.O.M. and Y.M. contributed to conceptualization, resources, investigation, supervision, and writing.

## Competing interests

The authors declare no competing interests.

## Additional information

[1]Environmental Change Institute, School of Geography and the Environment, University of Oxford, Oxford, United Kingdom. [2]Leverhulme Centre for Nature Recovery, University of Oxford, Oxford, United Kingdom. [3]Forestry Research Institute of Ghana, Council for Scientific and Industrial Research, Kumasi, Ghana. [4]Department of Natural Resources Management, CSIR College of Science and Technology, Kumasi, Ghana. [5]Centro Euro-Mediterraneo sui Cambiamenti Climatici, Leece, Italy. [6]Georgina Mace Centre for the Living Planet, Department of Life Sciences, Imperial College London, Silwood Park Campus, Buckhurst Road, Ascot, United Kingdom. [7]Forestry Division, Food and Agriculture Organization of the United Nations, Panama City, Panama. [8]Department of Forest Ecology, Faculty of Forestry and Wood Sciences, Czech University of Life Sciences, Praha, Czech Republic. [9]School of Biological Sciences, University of Adelaide, Adelaide, South Australia, Australia. [10]Groningen Institute for Evolutionary Life Sciences, University of Groningen, P.O. Box 11103, 9700 CC Groningen, The Netherlands. [11]Department of Earth System Science, Ministry of Education Key Laboratory for Earth System Modeling, Institute for Global Change Studies, Tsinghua University, Beijing, China. [12]AMAP (Botanique et Modelisation de l'Architecture des Plantes et des Végétations), CIRAD, CNRS, INRA, IRD,Université de Montpellier, Montpellier, France. ✉e-mail: huanyuan.zhang@ouce.ox.ac.uk; yadvinder.malhi@ouce.ox.ac.uk

