## [Peer Review File · Nature Communications]

Contrasting carbon cycle along tropical forest aridity gradients in West Africa and AmazoniaREVIEWER COMMENTS

Reviewer #1 (Remarks to the Author):

The paper presents a comparison of GPP and component fluxes across an aridity gradient in Amazonia and in W. Africa. The main insight is that the allocation to NPP:stem and NPP:canopy appears to be different on the two different continents, and CUE is not constant across environmental gradients, but for different reasons on each continent. They also compare these results to FLUXNET data and MODIS GPP data.

The results are interesting, and I do feel that this manuscript makes further insights from Moore et al GCB 2017 (which reported on patterns of NPP in W. Africa) and Malhi et al GCB 2015 (which looked at patterns of GPP, respiration and NPP in Amazonia). i.e. there is reason to publish this new comparison between the two already largely published datasets.

However, the writing could use some work in terms of clarity, and the authors could better justify their interpretations through carefully referring to particular results. In particular, none of the multi-panel plots are labelled so often it is not clear which component is being referred to when they mention "Figure 2" for example.

I do also question/disagree with some of their conclusions - see detailed comments below regarding their explanation for why CUE shows a unimodal relationship with aridity. I am sure that more clarity of writing and more careful referencing to specific results can clear this up.

It is interesting to see the Fluxnet and MODIS estimates but it would have been nice to also show what DGVM's predict - the TRENDY intercomparison has been published I believe, so this could also have been presented?

Finally some more biology would be nice - photographs of the sites, and some better justification for why you believe that the outlier BOB sites do actually represent all semi-deciduous forest ecosystems in Africa. The discussion on this was interesting but could have been expanded a little.

Overall very enjoyable.

Detailed comments:

line 120 - total NPP does not seem to be higher in WAfrica than Amazonia. If you believe so then can I see a stats test to prove it? Perhaps you just mean the NPP by litterfall, which does appear to be substantially higher in the BOB sites in W-Africa. Please provide more clarity here - if you label your subplots in your figures with a, b, c, d then we will know exactly which flux you are talking about.

Line 120 - I struggled to find table S2 - please put all tables together ?

Line 126 - "stem woody productivity "OF THE WETTEST SITES" and rhizosphere respiration

Line 143 - can you provide information on the seasonal aridity in Table S1? you keep distinguishing these forest types in terms of their seasonality of rainfall but you don't provide data showing us what this is? Is MCWD your metric of seasonal aridity? but this combines rainfall and seasonality? There are metrics that just present the distribution of the rainfall, independently of the amount?

Line 146 - I disagree (fine root productivity shows no pattern with rainfall, and leaf production shows a unimodal relationship only in Africa). The increase in CUE appears to occur for different reasons in Amazonia and W. Africa. In W. Africa it is due to high leaf production, which, even though leaf respiration also increases, appears to drive the increased NPP. In Amazonia leaf production is not higher at intermediate aridity sites, but leaf respiration decreases, which presumably causes the increased CUE at intermediate rainfall. I think highlighting the totally contradictory responses of leaf respiration in the two continents is important and interesting. Again, your interpretations would be easier to follow if you referred explicitly to different panels (a,b,c etc) in your text.

Line 151: "The pattern of GPP along the aridity gradient is well simulated in MODIS but not FLUXCOM (Figure 4)." Again, I do not understand how you come to this conclusion. Fluxcom appears to simulate the decline in GPP with aridity observed in Amazonia, and MODIS appears to simulate the peak of GPP in intermediate aridities observed in Africa - i.e. The models are unable to predict the contrasting responses to aridity shown on the two continents, and predict different responses.

Lin 154 - can you please be more explicit about how you conclude that arid forests are more productive in Amazonia? I don't see this pattern anywhere in the data. The only variable with a unimodal distribution in Amazonia is CUE?

Line 162 - NPP does appear to differ between these sites - just GPP doesn't!

Line 163 - do you mean Figure 4? Perhaps you mean Figure 3?

Line 163 "In contrast, previous studies in Amazonia reported that GPP and NPP change considerably from wet to dry sites 11,38,39 and along forest-savanna transition 40"
This sentence seems out of place? In this paragraph you appear to be discussing local variation among sites with the same rainfall? Overall this section is very hard to follow.

Line 169 - but if the errors were caused by observer bias then this could still be a factor? Presumably different teams were measuring the different sites.

Line 174 - repetition

Line 180-181- important, perhaps consider putting these figures in the main text.

Line 188- please label a and b in the figures

Line 203: "NPP partitioning to the canopy is the most dominant portion in our study site,"
What does this mean? Do you mean, the fraction of NPP allocated to canopy is the largest fraction? That does not appear to be true from Figure 3. Please expand and explain, it seems important.

Line 224 - well... could also be measurement error/observer bias right? A badly calibrated scale could artificially inflate all leaf litter measurements for example. But lines 226 - 227 are reassuring

Line 235: "Species at BOB and KOG were indeed characterized by high photosynthesis rates" can you name some of these species? Nice to bring some functional ecology into this

Line 236 - I know you reference studies that report photosynthetic rates, but I would have found this manuscript far easier to interpret if you presented some information on species-specific leaf-level photosynthetic rates for each site here as well.

Line 242: "with more seasonality" what do you mean by more seasonality? do you mean more seasonal leaf display, or more seasonal water availability? - do the trees at this site also show a semi-deciduous strategy with short leaf lifespan?

Line 250 - again, can you report on seasonality separately from MCWD? - do you have information on leaf lifespan for each site? If so can you report on that?

Line 258 - I don't think you can be subject to a hypothesis? You can be subject to intermediate disturbance, and, as predicted by the intermediate hypothesis, will therefore have high productivity and turnover.

Line 260 - on both continents? or are you just talking about Africa here? Please be specific about whether you are generalising or discussing the African tropical forest situation.

Line 787: "The following tests were performed as NPP were commonly calculated from GPP with a 787 fixed proportion (known as CUE). " what does this mean?

Overall: it appears that the patterns in Africa to a large extent are driven by the responses of BOPIRI - can you give more information about this site- its floristics for example?

Supplementary materials:

You highlight the importance of seasonality and leaf phenology in your discussion but you don't indicate how you reported and dealt with seasonal variation in LAI for your sites. Did it vary over the season in some sites? if so, how did you deal with that.

Sally Archibald

Reviewer #2 (Remarks to the Author):

This enlightening and thought-provoking study contributes to fill a very substantial data gap around the productivity of tropical forests. Whilst there is quite a lot of data on woody growth, NPP and GPP data is rare in the tropical forests. Not surprisingly, as it is extremely labour-intensive to collect. This rarity is, however, a major problem because both understanding how the forest functions and effectively assessing its carbon cycling requires that we have good quantifications of GPP and especially NPP. The authors found that GPP was much higher in the African tropical forests than previously thought and they highlight a major deviation between ground observations and the most commonly used large-scale products for looking at productivity in this region. It also highlights mechanistic reasoning underlying the higher productivity found in Africa.

The disconnect between woody NPP and overall NPP and GPP is particularly important information for piecing together how the allocation strategies of tropical forest trees vary in space. Allocation is one of the greatest uncertainties in modelling of the forest carbon cycle, with models realising drastically different patterns (e.g. Pugh et al., 2020, *Biogeosciences* 17, 3961-3989). Given the sourced-based nature of most of our efforts to model the future forest carbon cycle, information on NPP allocation (as opposed to the much more commonly found and less challenging biomass partitioning) as provided by this study is desperately needed. The results here both provide an important piece of the overall puzzle of African (and tropical) forest function and carbon cycling, and provide strong motivation to further investigate primary productivity and allocation across these forests. I cannot identify any major problems with the methodology for assembling the observations (I am not qualified to comment on the execution of the field measurement techniques themselves). I expect the results will be of wide interest across the community of scientists working on forest ecology and carbon cycling. I only have minor comments and very much recommend publication.

Minor comments

Figure 2. The relative ranking means that we cannot see the true shape of the relationships with CWD, which limits the extent to which it can be interpreted. It would be much more informative to plot this along an x-axis of absolute CWD.

Line 95-97. Satellite GPP estimates are also models, not observations. So the lack of evidence is much more than cloud cover. Emphasising that only models (MODIS-GPP, Fluxcom and indeed DGVMs) without ground truthing are being used to estimate carbon fluxes in African forests would strengthen your point of the need for field measurements.

135. This would be a much more powerful statement if you extracted the information from model simulations to compare.

181. Typo "misrepresent"

737. "using a previously-developed equation"

743. "allometric"

769. "GPP and CUE"

Response to reviewers' comments

Please note that your original comments are in green and our response is in black. A light green colour is chosen for visual impairment consideration.

REVIEWER COMMENTS

Reviewer #1 (Remarks to the Author):

The paper presents a comparison of GPP and component fluxes across an aridity gradient in Amazonia and in W. Africa. The main insight is that the allocation to NPP:stem and NPP:canopy appears to be different on the two different continents, and CUE is not constant across environmental gradients, but for different reasons on each continent. They also compare these results to FLUXNET data and MODIS GPP data.

The results are interesting, and I do feel that this manuscript makes further insights from Moore et al GCB 2017 (which reported on patterns of NPP in W. Africa) and Malhi et al GCB 2015 (which looked at patterns of GPP, respiration and NPP in Amazonia). i.e. there is reason to publish this new comparison between the two already largely published datasets.

Response: Thank you for your affirmation. This is exactly the key focus of the manuscript.

However, the writing could use some work in terms of clarity, and the authors could better justify their interpretations through carefully referring to particular results. In particular, none of the multi-panel plots are labelled so often it is not clear which component is being referred to when they mention "Figure 2" for example.

Response: Thanks for your suggestions. We have done so. Information like "Figure 2a" was also added to the main text.

I do also question/disagree with some of their conclusions - see detailed comments below regarding their explanation for why CUE shows a unimodal relationship with aridity. I am sure that more clarity of writing and more careful referencing to specific results can clear this up.

Response: We have rephrased them according to your suggestion, please see detailed response below.

It is interesting to see the Fluxnet and MODIS estimates but it would have been nice to also show what DGVM's predict - the TRENDY intercomparison has been published I believe, so this could also have been presented?

Response: You are absolutely right that a comparison with TRENDY would be interesting.

We currently are at an advanced stage of a separate manuscript that focuses on a data-model comparison (including TRENDY) that not only shows that models fail to capture the field observations, but then also explores in detail reasons why. We think a rich data-model comparison warrants a separate paper, and prefer to leave the DGVMs out of this paper. As a preview, the analysis below shows that DGVMs fail to capture field-measured GPP in Africa (the red dotted lines are GPP reported in this *Nature Communications* manuscript). In our follow-up paper, we explore why in detail.

The data-model comparison in fact has not been published anywhere (except for several Amazonia sites, see a study by other labs here <http://dx.doi.org/10.1088/1748-9326/aabc61>).

Confidential Response (please redact):

Confidential Response end

Finally some more biology would be nice - photographs of the sites, and some better justification for why you believe that the outlier BOB sites do actually represent all semi-deciduous forest ecosystems in Africa. The discussion on this was interesting but could have been expanded a little.

Response: Yes totally agree with you, fieldwork based manuscript is a lot more appealing with field photos! Many photos are added at the end of the supplementary 5.

Regarding the conclusion for productivity (GPP and NPP), as noted by the reviewer in later comments, we have discussed that this was not likely to be caused by sampling bias (i.e. happening to choose an abnormal site). But the reviewer is right that we did not discuss how representative these sites are. As this is a pioneering study of African forests' carbon budget, it is difficult to place in continent-wide context . BOB is unlikely to be representative of ALL semi-deciduous forests in Africa. Nonetheless, based on an endemic species map

(<http://dx.doi.org/10.2307/2258816>), we presume that ANK BOB and KOG are a reasonably representative choice of study site for West African semi-deciduous forests. The above is now added to Method and Discussion. We have been careful to ensure our writing does not prematurely claim this pattern is representative of all African tropical forests.

New Line 353:

As the study sites are situated in different forest zones based on specific endemism, we believe that ANK BOB and KOG are representative study sites of West African forests but they may not represent other African regions⁸³.

New Line 272:

Our analysis highlights that the studied West African forests have a contrasting carbon strategy to the studied Amazonian forests. However, as our African sites are restricted to one country, more evidence is required to achieve a more geographically distributed understanding of the African forest carbon cycle (and indeed of the Amazonian forest carbon cycle)

Overall very enjoyable.

Response: Thank you.

Detailed comments:

line 120 - total NPP does not seem to be higher in WAfrica than Amazonia. If you believe so then can I see a stats test to prove it? Perhaps you just mean the NPP by litterfall, which does appear to be substantially higher in the BOB sites in W-Africa. Please provide more clarity here - if you label your subplots in your figures with a, b, c, d then we will know exactly which flux you are talking about.

Line 120 - I struggled to find table S2 - please put all tables together ?

Response: Sorry for the confusion, now we have labelled figure 2. The sentence was referring to Table S2, where mean NPP of the W African sites is at 14.87 MgC/ha/year while mean NPP of the Amazonian sites is 12.86 MgC/ha/year, the z-test p-value for the difference of which was 0.05. However, we realised that at this place t-test is more appropriate so we replaced z-test in Table S2 with t-test, which has a p-value at 0.06. This is just marginally significant. We now replaced 'total NPP' with 'total GPP' so that the statement is well-founded.

Line 126 - "stem woody productivity "OF THE WETTEST SITES" and rhizosphere respiration

Response: Thank you we have done so.

Line 143 - can you provide information on the seasonal aridity in Table S1? you keep distinguishing these forest types in terms of their seasonality of rainfall but you don't provide data showing us what this is? Is MCWD your metric of seasonal aridity? but this combines rainfall and seasonality? There are metrics that just present the distribution of the rainfall, independently of the amount?

Response: Yes, very good point. We have calculated a rainfall **seasonality index** using equation 1 in this paper <https://doi.org/10.1007/s11269-022-03320-z>. The seasonality index is now presented in table S1.

New Line 349: Along the aridity gradient, rainfall seasonality (Table S1), vegetation seasonality and deciduousness increased considerably toward the drier sites ^{10,60,64}, whereas LAI and tree density decreased toward the drier sites ⁶³.

Line 146 - I disagree (fine root productivity shows no pattern with rainfall, and leaf production shows a unimodal relationship only in Africa). The increase in CUE appears to occur for different reasons in Amazonia and W. Africa. In W. Africa it is due to high leaf production, which, even though leaf respiration also increases, appears to drive the increased NPP. In Amazonia leaf production is not higher at intermediate aridity sites, but leaf respiration decreases, which presumably causes the increased CUE at intermediate rainfall. I think highlighting the totally contradictory responses of leaf respiration in the two continents is important and interesting. Again, your interpretations would be easier to follow if you referred explicitly to different panels (a,b,c etc) in your text.

Response: We have rephrased this according to your suggestion.

New Line 146: Along both transects, CUE peaks in the mid-aridity plots but this peak appears to occur for different reasons in Amazonia and West Africa. In Amazonia, leaf production is not higher, but leaf respiration is lowest at intermediate aridity sites leading to the high CUE. In West Africa, leaf production is so high at mid-aridity site that, even though leaf respiration also increases, drives the high CUE at the mid-aridity site. Woody stem productivity does show strong site-to-site variation, but no clear pattern along both aridity gradients. Surprisingly, along the African transect, GPP, autotrophic respiration, CUE and NPP are highest in the mid-aridity site.

Line 151: "The pattern of GPP along the aridity gradient is well simulated in MODIS but not FLUXCOM (Figure 4)." Again, I do not understand how you come to this conclusion. Fluxcom appears to simulate the decline in GPP with aridity observed in Amazonia, and

MODIS appears to simulate the peak of GPP in intermediate aridities observed in Africa - i.e. The models are unable to predict the contrasting responses to aridity shown on the two continents, and predict different responses.

Response: Yes the models are unable to predict the contrasting pattern of GPP. We have rephrased this paragraph according to your suggestion.

New Line 158: The contrasting patterns of GPP along both aridity gradients are not simulated by MODIS nor FLUXCOM (Figure 4). MODIS simulated that tropical GPP should peak at mid-aridity sites, while FLUXCOM believed that tropical GPP decreased toward drier sites.

Lin 154 - can you please be more explicit about how you conclude that arid forests are more productive in Amazonia? I don't see this pattern anywhere in the data. The only variable with a unimodal distribution in Amazonia is CUE?

Response: Sorry, this is a wording issue. It is comparing the West African mid-arid site (BOB) to wet forests (ANK), and comparing BOB to the Amazonian lowland forests. We have also changed the wording from 'Arid forests BOB' to 'semi-deciduous forests BOB' because BOB is not a dry forest (KOG02 is more appropriate to be called dry forest) and it is more important to highlight 'semideciduous'

New Line 155: This study therefore highlights that semideciduous forests with intermediate aridity can be more productive than wet evergreen forests, as seen in comparing the mid-aridity site (BOB) to wet sites (ANK and ALP) (Figure 2a and 2c).

Line 162 - NPP does appear to differ between these sites - just GPP doesn't!

Response: Sorry to omit this, we have rephrased accordingly.

New Line 168: The steep variation of biomass along forest-savanna transition in Kogyae(KOG) leads to variation in NPP but not GPP.

Line 163 - do you mean Figure 4? Perhaps you mean Figure 3?

Response: Sorry for the mistakes, it should be Figure 3. We have removed this sentence because the statement does not hold very well in Figure 3.

Line 163 "In contrast, previous studies in Amazonia reported that GPP and NPP change considerably from wet to dry sites 11,38,39 and along forest-savanna transition 40"
This sentence seems out of place? In this paragraph you appear to be discussing local variation among sites with the same rainfall? Overall this section is very hard to follow.

Response: Yes the main purpose of the paragraph is to discuss local variation (plot-to-plot variation within a site), and compare plot-to-plot variation to site-to-site variation (a study site contains several on hectare plots). The main point is that site-to-site variation (along aridity gradient) is a lot more prominent. We have rewritten this paragraph to make the point clear.

Line 161: Within-site variation (i.e. adjacent plot-to-plot variation, not site-to-site variation discussed above) can reveal the impact of local environmental factors. The seasonally flooded swamp forest ANK-03, despite having less stem biomass (135 MgC/ha) than the adjacent dry substrate forest (163 and 153 MgC/ha for ANK-01 and ANK-02), does not have significantly different NPP or GPP (Figure S 4). Similarly, the different history of logging in Bobiri (see Methods) do not seem to result in significant difference in either NPP or GPP (except BOB-02, Figure S 4). The steep variation of biomass along forest-savanna transition in Kogyae(KOG) leads to variation in NPP but not GPP. Similarly, GPP and NPP vary considerably along continental aridity gradient in Amazonia but variation amongst plots within the same site is mild ^{11,40-42}. Plot-to-plot variation of NPP with logging intensity was reported in Borneo ⁴³, but that was for much higher intensities of logging than at the Bobiri site. Such plots-to-plot variation on GPP and NPP were not seen for our Bobiri study and another West-African study on logging ⁴⁴. The above implies that aridity, instead of local edaphic or other environmental factors, is the primary driver of productivity of West African forests. The lack of plot-to-plot variation within a site also increases confidence in the exceptionally high GPP measurements found in Bobiri.

Line 169 - but if the errors were caused by observer bias then this could still be a factor? Presumably different teams were measuring the different sites.

Response: Yes this is discussed in a later paragraph, see response to your later comments.

Line 174 - repetition

Response: Sorry we have corrected this

Line 180-181- important, perhaps consider putting these figures in the main text.

Response: Thank you for your suggestion, are you referring to Figure S1 at line 178? We put it in supplementary since the figure is a large panel and we only made one statement here referring to it. We are happy to move it to the main text if the editor finds it fits into the main text space.

Line 188- please label a and b in the figures

Response: Apologies, we have done so

Line 203: "NPP partitioning to the canopy is the most dominant portion in our study site," What does this mean? Do you mean, the fraction of NPP allocated to canopy is the largest fraction? That does not appear to be true from Figure 3. Please expand and explain, it seems important.

Response: Yes we are comparing the fraction of NPP canopy allocation to other organs. Thanks for pointing this out, we overlooked the calculation of mean stats for allocation, we have added a new Table S3 to provide stats. Indeed, on average, more NPP was allocated to canopy than woody allocation. Probably figure 3 is not very intuitive because the grid lines are not coloured. We have coloured them to help readers.

We noticed that the statement at line 197: "Most variation can be seen in photosynthate allocated to the canopy, with a canopy respiration range from ..." is weak. We have rewritten this part to explain that both canopy and woody allocation varied a lot spatially.

Line 206: The high variability in allocation patterns raises questions about the "fixed-ratio method" for intact forests carbon modelling^{53,54}. Despite the large spatial variation, on average, tropical NPP partitioning to the canopy is slightly larger than to wood (Table S3), a pattern not captured by satellite-based products or vegetation models which substantially underestimate allocation to canopy^{52,55}. Moreover, along both the Amazonia and West African aridity gradients, spatial patterns in stem NPP are poor analogues for the spatial pattern of total NPP or GPP – hence inferences on tropical forest productivity based on forest censuses alone should be treated with caution (Figure 2i, 2a). Previous studies also found canopy NPP could explain greater spatial variation of total NPP compared with stem NPP^{56,57}. The gathered evidence highlights the great importance of CUE and canopy NPP, whereas previous literature has paid much more attention to GPP and stem NPP⁵⁸; hence, future research is needed to balance and complete the carbon cycle.

Line 224 - well... could also be measurement error/observer bias right? A badly calibrated scale could artificially inflate all leaf litter measurements for example. But lines 226 - 227 are reassuring

Response: Thank you.

Line 235: "Species at BOB and KOG were indeed characterised by high photosynthesis rates" can you name some of these species? Nice to bring some functional ecology into this

Line 236 - I know you reference studies that report photosynthetic rates, but I would have found this manuscript far easier to interpret if you presented some information on species-specific leaf-level photosynthetic rates for each site here as well.

Overall: it appears that the patterns in Africa to a large extent are driven by the responses of BOPIRI - can you give more information about this site- its floristics for example?

Response: Your comments above are grouped since these very constructive points all pertain to floristics and photosynthesis traits. To address your comments, we have added photosynthetic traits measurement (net assimilation rate at 400 ppm under light saturation, Asat; net assimilation rate at 2000 ppm under light saturation, Amax) to Table S1 (reported in previous studies). We have also added table S 4 to list common species at each plot and their Asat Amax measurements.

Line 242: Such a carbon strategy, however, entails higher leaf-level photosynthesis to provide more carbon replacing leaves lost via turnover⁶¹. Species at BOB and KOG were indeed characterized by high photosynthesis rates^{29,30}, for example, *Triplochiton scleroxylon* (BOB), *Nesogordonia papaverifera* (BOB), *Azelia Africana* (KOG) and *Pterocarpus erinaceus* (KOG) (all deciduous) (Table S4).

Line 242: "with more seasonality" what do you mean by more seasonality? do you mean more seasonal leaf display, or more seasonal water availability? - do the trees at this site also show a semi-deciduous strategy with short leaf lifespan?

Line 250 - again, can you report on seasonality separately from MCWD? - do you have information on leaf lifespan for each site? If so can you report on that?

Response for the above two comments: Yes we have added rainfall seasonality per your previous comments. Lifespan of some of the sites have been published so we have added lifespan to table S1 by parsing data from this paper <https://doi.org/10.5194/bg-2019-175>. Yes for deciduous strategy with short leaf lifespan.

In fact both rainfall and leaf display vary seasonally. Here we were talking about leaves. We have made this clear and cited the papers on seasonality along Amazonia aridity gradient for readers' information.

New Line 250: Similar to BOB02, another mid-aridity plot in Amazonia, Kenia (KEN01), with the highest NPP along the Amazon aridity gradient, also featured a similar carbon strategy with more canopy seasonality, higher CUE, and quicker carbon turnover (see details of Kenia site⁴¹ and leaf lifespan and seasonality along the aridity gradient^{60,64}).

Line 258 - I don't think you can be subject to a hypothesis? You can be subject to intermediate disturbance, and, as predicted by the intermediate hypothesis, will therefore have high productivity and turnover.

Response: very good point, we have rephrased per your suggestion.

Line 260 - on both continents? or are you just talking about Africa here? Please be specific about whether you are generalising or discussing the African tropical forest situation.

Response: Sorry for the confusion, here we are talking about West African semi-arid forests compared to Amazonia wet evergreen. We have specified so. Not sure whether this pattern could be generalised to 'Tropical seasonally dry forests are more resilient to drought than wet-evergreen', given mixed evidence of a meta-analysis

(<https://iopscience.iop.org/article/10.1088/1748-9326/aa5968>) and a Costa Rica study (<https://onlinelibrary.wiley.com/doi/full/10.1111/j.1365-2486.2010.02326.x>). The above two papers are not cited and discussed. We believe that here we should not overly generalise, but stick with the results presented in this manuscript.

New line 270: Previous studies reveal that long-term drying trends have driven West African semi-arid forests toward more deciduousness^{6,74}, perhaps making them more resilient to climate change than wet-evergreen Amazonia forests^{5,7}.

Line 787: "The following tests were performed as NPP were commonly calculated from GPP with a 787 fixed proportion (known as CUE). " what does this mean?

Response: Sorry for the confusion, here it is meant to say: CUE is calculated as NPP/GPP. We have rephrased this.

Supplementary materials:

You highlight the importance of seasonality and leaf phenology in your discussion but you don't indicate how you reported and dealt with seasonal variation in LAI for your sites. Did it vary over the season in some sites? if so, how did you deal with that.

Response: Thank you for pointing this important omission out. Indeed, many components, including LAI, have strong seasonal variation (<https://doi.org/10.1002/2015GB005270>). Since this paper focuses on the spatial variation of the carbon budget, not seasonal variation nor long-term spatial variation, here we average monthly measurements to an 'annual mean' for both study gradients, which represents total productivity of an ecosystem. This is actually quite important information; we have added this to Method:

Line 397: Since this paper focuses on the spatial variation of carbon budget, not seasonal variation (presented here⁶⁴), we average monthly measurements to an 'annual mean' for both study gradients.

Reviewer #2 (Remarks to the Author):

This enlightening and thought-provoking study contributes to fill a very substantial data gap around the productivity of tropical forests. Whilst there is quite a lot of data on woody growth, NPP and GPP data is rare in the tropical forests. Not surprisingly, as it is extremely labour-intensive to collect. This rarity is, however, a major problem because both understanding how the forest functions and effectively assessing its carbon cycling requires that we have good quantifications of GPP and especially NPP. The authors found that GPP was much higher in the African tropical forests than previously thought and they highlight a major deviation between ground observations and the most commonly used large-scale products for looking at productivity in this region. It also highlights mechanistic reasoning underlying the higher productivity found in Africa.

Response: Thank you

The disconnect between woody NPP and overall NPP and GPP is particularly important information for piecing together how the allocation strategies of tropical forest trees vary in space. Allocation is one of the greatest uncertainties in modelling of the forest carbon cycle, with models realising drastically different patterns (e.g. Pugh et al., 2020, *Biogeosciences* 17, 3961-3989).

Response: You have a very good point here! Now discussed at Line 210:

New Line 210: The high variability in allocation patterns raises questions about the “fixed-ratio method” for intact forests carbon modelling^{54,55}. Dynamic NPP allocation to some vegetation components are considered by current models but with models realising drastically different patterns²⁹. Despite the large spatial variation, on average,

Given the sourced-based nature of most of our efforts to model the future forest carbon cycle, information on NPP allocation (as opposed to the much more commonly found and less challenging biomass partitioning) as provided by this study is desperately needed. The results here both provide an important piece of the overall puzzle of African (and tropical) forest function and carbon cycling, and provide strong motivation to further investigate primary productivity and allocation across these forests. I cannot identify any major problems with the methodology for assembling the observations (I am not qualified to comment on the execution of the field measurement techniques themselves). I expect the results will be of wide interest across the community of scientists working on forest ecology and carbon cycling. I only have minor comments and very much recommend publication.

Response: Thank you so much for your affirmation.

Minor comments

Figure 2. The relative ranking means that we cannot see the true shape of the relationships with CWD, which limits the extent to which it can be interpreted. It would be much more informative to plot this along an x-axis of absolute CWD.

Response: Thank you very much for your suggestion. We tried to plot figure 2 along absolute MCWD as provided in Table S1. However, since many plots share the same MCWD values, these plots overlap with each other (see below, should we put this in supplementary? Please let us know). Plotting along an x-axis of surface volumetric water contents could distinguish each plot but this index is more associated with local topography and soil property which fails to reflect the variation of precipitation from site to site. To address your comment, we have now added MCWD values to x-axis in the main figure for reader's information for now (please check the updated figure 2 in the main text).

Line 95-97. Satellite GPP estimates are also models, not observations. So the lack of evidence is much more than cloud cover. Emphasising that only models (MODIS-GPP, Fluxcom and indeed DGVMs) without ground truthing are being used to estimate carbon fluxes in African forests would strengthen your point of the need for field measurements.

Response: thank you for your suggestion, we have added a sentence here.

Line 97: Satellite-based GPP products (for example MODIS-GPP) struggle to provide reliable estimates of GPP of the region owing to the dense cloud cover, which extends across both wet and dry seasons²¹⁻²³. Further, satellite based GPP products also require vegetation process modelling that needs to be informed by field measurements.

135. This would be a much more powerful statement if you extracted the information from model simulations to compare.

Response: You are absolutely right that a comparison with models would be interesting.

We currently are at an advanced stage of a separate manuscript that focuses on a data-model comparison (including TRENDY) that not only shows that models fail to capture the field observations, but then also explores in detail reasons why. We think a rich data-model comparison warrants a separate paper, and prefer to leave the DGVMs out of this paper. As a preview, the analysis below shows that DGVMs fail to capture field-measured GPP in Africa (the red dotted lines are GPP reported in this *Nature Communications* manuscript). In our follow-up paper, we explore why in detail.

The data-model comparison for several Amazonia sites, see a study by other labs here <http://dx.doi.org/10.1088/1748-9326/aabc61>. We have modified the main text to make this point stronger:

Line 141: We found FLUXCOM and MODIS consistently underestimated both Amazonia and West African forests GPP (Figure 4), a finding consistent with previous Amazonia data-model comparison studies^{36,37}. There was only one eddy covariance tower for West African forests, situated at one of our study sites Ankasa (ANK)^{19,20}. Previous studies found that, at this site, the flux-tower GPP (yearly mean varying from 22 to 36 MgC/ha/year) is larger than vegetation model simulation³⁸⁻⁴⁰, but both are substantially smaller than in-situ biometric measurements (40.1 MgC/ha/year)³⁸⁻⁴⁰.

Confidential Response (please redact):

Confidential Response end

181. Typo “misrepresent”

Response: Sorry for the typo, we have corrected it.

737. “using a previously-developed equation”

Response: Thank you, we have corrected this.

743. “Allometric”

Response: Apologies, we have corrected it.

769. “GPP and CUE”

Response: Sorry for this omission, we have corrected it.

REVIEWER COMMENTS

Reviewer #2 (Remarks to the Author):

I find the authors have done a great job of addressing the very minor points raised. As before, I recommend publication. I'm also looking forward to the follow-up paper unpicking the DGVM responses (and thanks for the sneak preview!)

The version of Figure 2 in the response document which is plotted directly against MCWD seems to have some strange shapes driven by the function which is chosen to fit (peaks which are not directly supported by any datapoint for the West African dataset). I agree with the authors that the new Fig 2 in the main text with the MCWD values on the x-axis is the better compromise.